# Catechins as a Potential Dietary Supplementation in Prevention of Comorbidities Linked with Down Syndrome

**DOI:** 10.3390/nu14102039

**Published:** 2022-05-12

**Authors:** Christophe Noll, Janany Kandiah, Gautier Moroy, Yuchen Gu, Julien Dairou, Nathalie Janel

**Affiliations:** 1Division of Endocrinology, Department of Medicine, Centre de Recherche du CHUS, Université de Sherbrooke, Sherbrooke, QC J1K 2R1, Canada; christophe.noll@usherbrooke.ca; 2Unité de Biologie Fonctionnelle et Adaptative, UMR 8251 CNRS, Université Paris Cité, F-75013 Paris, France; kd.janany@gmail.com (J.K.); guyuchen0509@hotmail.com (Y.G.); 3Unité de Biologie Fonctionnelle et Adaptative, INSERM CNRS, Université Paris Cité, F-75013 Paris, France; gautier.moroy@u-paris.fr; 4Laboratoire de Chimie et Biochimie Pharmacologiques et Toxicologiques, UMR 8601 CNRS, Université Paris Cité, F-75006 Paris, France; julien.dairou@u-paris.fr

**Keywords:** catechins, trisomy 21, Alzheimer’s disease, metabolic syndrome, Dyrk1A

## Abstract

Plant-derived polyphenols flavonoids are increasingly being recognized for their medicinal potential. These bioactive compounds derived from plants are gaining more interest in ameliorating adverse health risks because of their low toxicity and few side effects. Among them, therapeutic approaches demonstrated the efficacy of catechins, a major group of flavonoids, in reverting several aspects of Down syndrome, the most common genomic disorder that causes intellectual disability. Down syndrome is characterized by increased incidence of developing Alzheimer’s disease, obesity, and subsequent metabolic disorders. In this focused review, we examine the main effects of catechins on comorbidities linked with Down syndrome. We also provide evidence of catechin effects on DYRK1A, a dosage-sensitive gene encoding a protein kinase involved in brain defects and metabolic disease associated with Down syndrome.

## 1. Introduction

### 1.1. Polyphenols and Health Promoting Effects

Polyphenols are plant metabolites, termed from their chemical structures of repeating phenolic moieties with/without other functional groups [1]. Over several hundred polyphenols have been discovered in fruits, vegetables, nuts, and seeds.

They are categorized as non-flavonoids and flavonoids, the flavonoids being characterized by the presence of a flavan nucleus, which is composed of at least 15 carbon atoms with two atomic rings (Figure 1A). More than 7000 flavonoids have been identified in natural sources.

Flavonoids are categorized into six major subclasses [2]. Once ingested, flavonoids undergo extensive metabolism in the small and large intestines, in the liver, and within cells [3]. They are converted into an aglycone form, these aglycones being conjugated into O-glucuronides or O-sulphates in the liver by xenobiotics metabolizing enzymes and transported into the bloodstream until urinary excretion [4].

Research regarding polyphenols has gained prominence over the years because of their potential as pharmacological nutrients with many potential health-promoting effects [5]. Regarding their potential antioxidant activity, many studies have considerably demonstrated the role of total flavonoids and specific subclasses in treating various diseases and attenuating the risk factors for chronic diseases, mainly diabetes, cardiovascular disease (CVD), cognitive disorders and all-cause mortality [6,7,8].

Cognitive capacities can be improved by daily flavonoid consumption and, more precisely in older adults, these results raise the perspective of protection against neurodegenerative diseases, such as Alzheimer’s disease (AD) [9,10,11]. Indeed, they exert neuroprotective effects by inhibiting oxidative stress, neuroinflammation and key enzymes forming amyloid plaques and other toxic products [12].

### 1.2. Catechins, Oxidative Stress and Inflammation

Most polyphenols are flavanols, commonly known as catechins, and are promising candidates in the field of biomedicine. They have many chemical structural features, such as hydroxyl groups (−OH), combining easily with other materials. There are eight catechins: Catechin (C), EpiCatechin Gallate (ECG), EpiGalloCatechin (EGC), EpiCatechin (EC), EpiGalloCatechin Gallate (EGCG), GalloCatechins (GC), Catechin Gallate (CG), and GalloCatechin Gallate (GCG) (Figure 1B) [13].

Catechins are widely distributed in many foods and herbs [14]. EC is mainly present in apples, red fruits, broad beans, pears, chocolate and cacao products, C, EC, and GC are present in red wine, where their reaction with tannins is responsible for wine flavor. Catechins are one of the main active biological ingredients of tea leaves, accounting for more than 75% of the polyphenol compounds, EGCG (an ester of EpiGalloCatechin and Gallic Acid) being predominant.

The imbalance between reactive oxygen species (ROS) and antioxidant molecules causes oxidative stress and is implicated in inflammation. CVD, diabetes, and neurodegenerative diseases are associated with oxidative stress and inflammation [15]. Due to their chemical structure and the total number of phenolic hydroxyl groups in polyphenols [16], catechins possess direct and indirect antioxidant effects. Phenolic hydroxyl groups can react with ROS and reactive nitrogen species reducing free radicals [17,18]. They also regulate the synthesis of proteins, such as catalase (CAT), glutathione (GSH), superoxide dismutase (SOD), and nicotinamide adenine dinucleotide phosphate (NADPH), molecules implicated in the maintenance of redox balance [19,20]. The most effective compounds in the protection against ROS-induced erythrocyte hemolysis are C and EC, while ECG, EGCG and EGC protected at the lowest concentrations against hypochlorite-induced hemolysis [21]. EGCG and ECG are highly effective free-radical scavengers [22].

At the same time, as indirect antioxidant mechanisms, catechins regulate protein activities and signaling strategies, inducing antioxidant enzymes, inhibiting pro-oxidant enzymes, inducing phase II metabolizing enzymes, depending on concentrations [16,23]. EGCG and its derivatives are major bioactive components with potent antioxidant properties [24]. They can regulate cell survival molecules, caspases, and NADPH oxidase (NOXs), resulting in the decrease in superoxide production and the transcription factor nuclear factor (NF)-κB activity [16,23,25,26]. Synergistic effects with GSH need the presence of a catechol group for antioxidant capacity [27].

In response to oxidative stress, the defense mechanisms will be carried out with an increase in enzymatic antioxidants and other low molecular weight antioxidants [28]. Piperine, which can raise the EGCG bioavailability, has a positive antioxidative potential consecutively on some antioxidant enzymes in combination with EGCG [28]. EGCG and EC can increase catalase activity through decreasing ROS [29,30].

Recent studies have focused on maximizing the efficacy of antioxidants. Enzymatic glucosylation of EGCG and lipophilic EGCG derivatives have shown increased antioxidant activity in different cellular models [31,32,33]. In addition to ROS-related mechanisms, EGCG–protein interactions can explain the health beneficial effects [34].

Catechins may act as a therapeutic agent by suppressing many oxidative stress-related pathways responsible for the inflammation processes [35,36] by modulating NF-κB activity and activator protein-1 (AP-1) [37]. Catechins can inhibit the oxidative damage and inflammation through the crosstalk between the MAPKs and NF-κB pathways, thus demonstrating anti-inflammatory effects [38]. Inhibition of oxidative stress and inflammation can be also due to Nrf2 activation [24]. ECG might inhibit the p38, JNK, phosphatidylinositol-3 kinase/Akt (PI3k/Akt) pathways and NF-κB downstream pathway, and at the same time can enhance Nrf2 activation, resulting in a significant increase in GSH and Heme oxygenase 1 (HO-1) expression [39]. EGCG could also inhibit the secretion of inflammatory cytokines by attenuating the NF-κB pathway [28].

One major limitation of catechins is low bioavailability. Catechins can be absorbed by the gastrointestinal tract. They may undergo three types of metabolic pathways: methylation, glucuronidation, and sulfonation [40]. The only flavonol form present in a significant plasma percentage is EGCG [41], the others being detected as glucoronidated or sulphated forms [40]. Metabolization leads to metabolites that can extend the beneficial effect of catechins, having a longer half-time. Maintaining or increasing stability and bioavailability of these molecules within living organisms may be required for some protective mechanisms.

EGCG and ECG have shown anti-obesity, anti-diabetic effects, and focused on the use on neurodegenerative disorders, such as AD. Many studies have analyzed EGCG effects on several aspects of Down syndrome (DS), the most common genetic disorder that causes intellectual disability and that is characterized by increased incidence of AD, obesity, and metabolic disorders. Therefore, it will be useful to see their effects on comorbidities linked with DS (Figure 2).

## 2. Down Syndrome and Catechins

### 2.1. Down Syndrome

DS, the most common chromosome abnormality among newborns (trisomy 21), has an estimated prevalence of 1 in 800 live births [42]. It is therefore the most common genetic developmental disorder. It must be noted that these numbers might be underestimated seeing the high probability of miscarriages and the advancement in medical concerning the prenatal diagnosis of DS fetuses allowing parents to make their own decision regarding pregnancy continuation [43]. Initially described by John Langdon Down in 1866 [44] it was studies led by Lejeune in 1959 that allowed the genetic comprehension cause of DS [45]. Genetically speaking, DS is caused by the triplication of the human chromosome 21 (HSA 21) entirely or partially and depending on how and when these chromosomic aberrations occur, they are several forms of DS. The most frequent chromosome errors are mostly due to nondisjunction errors of chromosome 21 during maternal meiosis resulting in the presence of an extra copy of all the chromosome 21 (karyotype 47, XX/XY, +21). Another chromosome aberration to a lesser extent is the translocation of an additional chromosome 21 to another chromosome (chromosome 14 usually) leading to a partial trisomy of HSA 21. It has been well-recognized that the main risk factor is the maternal age of pregnancy, especially during maternal meiosis [46].

Phenotypical characteristics are variable and complex, reduced learning and memory capacities being common to all individuals with DS, the delay in the cognitive development leads to intellectual disability [47]. The intellectual disability, together with other neurological features, such as learning disability and weak working memory, is one of the well-known clinical characteristics of DS due to altered brain development. At birth, DS is recognizable by common physical features, such as muscular hypotonia, a small chin, slanted eye, a flat nasal bridge, a single crease on the palm and a protruding tongue due to a small mouth and a large tongue [43]. Usually, DS-associated clinical complications include Alzheimer-like disease, congenital heart disease, leukemia, hypertension, gastrointestinal problems, cancers, etc. [48]. DS is therefore a multiorgan disorder with immune and endocrine abnormalities, neurodegeneration begins later in life with AD-like neuropathological lesions [43]. Insulin resistance is a pathological common feature of type 2 diabetes (T2D), brain disorders, such as AD, and neurodevelopmental disorders, such as DS, contributing to their pathogenesis [49,50]. However, it must be noted that not all DS individuals are affected by the same medical problems and at same degree, it depends on the severity of DS in individual as well as their medical care during their lifespan. Despite these health issues, the average life expectancy of DS is around 60 years old, better than 50 years ago.

Research over the years demonstrate that triplication of HSA21 generates an imbalance in the expression of some gene located on this chromosome and likely causes disturbance on development and function at the origin of the phenotypes. Since the 1990s, to understand the molecular origin of some DS features, researchers study the molecular mapping of DS features by analyzing the correlation between genotype-phenotype in patients with partial trisomy 21 [51]. As followed, they delimited in 21q22 a “critical region” responsible for several phenotypes of DS. They suggested that this Down Syndrome Critical Region (DSCR) contains the dosage-sensitive genes that can explain the features observed in DS individuals. However, further analyses that have been performed in mice contest the implication of only one critical region [52,53]. Based on thorough analyses on DS mouse models, studies have attested that trisomy of DSCR is necessary but not sufficient to result in all DS clinical characteristics. In fact, by the presence of orthologous segments of HSA21 in mouse chromosomes Mmu10, Mmu16, and Mmu17, the creation of many genetically modified mouse models of T21, such as Dp(16)1Yey, Ts65Dn and TgDyrk1A by engineering have been possible [54]. They have been very helpful in many ways, such as the identification of dosage-sensitive genes, the understanding of the consequence on gene expression, and the understanding of the molecular mechanisms of pathogenesis-related to DS [55].

As follows, experimentations in mice having the expression of some genes increased or decreased allowed the highlighting of DYRK1A (dual specificity tyrosine phosphorylation-regulated kinase 1A) to be a candidate gene for intellectual deficiency, the most disabling phenotype in DS. It is suggested that this serine/threonine kinase DYRK1A localized in the DSCR has a major role in the central nervous system, thereby it is supposed to have an implication in cognitive impairment phenotype in DS individuals [56,57,58] as well as a skeletal deficit. DYRK1A has a major role in neurogenesis, neuronal differentiation, cell death, and synaptic plasticity [59]. This protein therefore plays a major role in cognitive dysfunctions linked with DS. Indeed, DYRK1A imbalance impacts the brain development, including pathways implicated in neurogenesis/neuroplasticity as well as synaptogenesis/synaptoplasticity [60,61]. More and more studies in mice focus their research on normalizing DYRK1A expression levels for therapeutic approaches for its multitude of interactions with many substrates (dynamin1, amphiphysin, tau protein, etc.) involved in cell cycle regulation and neuronal synaptic plasticity at different steps of development [62].

At present DS pathogenesis has not been fully explained, thus there is no treatment that allows full recovery from DS or impedes its occurrence. Treatments are only symptomatic and continuous medical care during the life of DS individuals is extremely important to impede or recline complications of any health issue [48].

### 2.2. Catechins in Molecular Effects of Down Syndrome

During the last years, numerous reports conducted on DS model mice and DS subjects have described catechins to be a major therapy for their beneficial effects in several aspects of DS.

ROS plays a key role in DS pathogenesis, DS patients displaying a pro-oxidant status [63]. Disturbance in redox homeostasis has been demonstrated in DS [64]. ROS production can be promoted by different genes mapping on has21, such as SOD1, RCAN1 and APP genes, causing dysfunction through the deregulation of signaling pathways, including AMPK, and PKA [65,66]. In lymphoblastoid cells and fibroblasts from DS subjects, EGCG modulates the cAMP/PKA signaling [67]. In neural hippocampal progenitor cells from the Ts65Dn mice, a mouse model of DS with three copies of most of the genes on MMU16, EGCG activates AMPK, correlating with the rescue of hippocampal neurogenesis [68]. EGCG at 20 mg/kg increased neurogenesis in adult hippocampal euploid mice by promoting the proliferation of neural progenitor cells in vitro and in vivo and ameliorated spatial learning and memory performance [69]. Using iPSCs from fetal fibroblasts of twins differing by their karyotypes, the authors strongly provide evidence of DYRK1A being responsible for neurogenesis defects in NPCs and neurons observed in DS cells. They demonstrate that the neural induction of the iPSCs into NPCs in presence of EGCG almost normalizes DYRK1A activity to value of control in these cells and rescues the proliferative ability as well as the apoptosis events, promoting neurogenesis. Corresponding effects are seen using DS-iPSCs with short hairpin RNA against DYRK1A [70].

Using transgenic mice overexpressing Dyrk1A, EGCG has been demonstrated as being able to cross the blood-brain barrier without adverse effects on cardiac and hepatic functions [71]. Consistently with previous findings, De La Torre et al. reported that one month of EGCG treatment in drinking water with Mega Green Tea Extract at 2–3 mg/day in young adult Ts65Dn and mice overexpressing Dyrk1A (TgDyrk1a) improved hippocampal-dependent learning deficits through the correction of DYRK1A overexpression [72]. Combined treatment of EGCG at an average dose of 30 mg/kg/day and environmental enrichment beneficially improved corticohippocampal-dependent learning and memory due to rescue of hippocampal dendritic spine density and preventing cognitive degeneration in Ts65Dn mice [73,74]. Ts65Dn mice treated with pure EGCG at 25 mg/Kg/day from 3 to 15 postnatal days have been observed [75] and exhibited the restoration of hippocampal neurogenesis and synaptic processes in the dentate gyrus (DG), neocortex and hippocampus. However, the beneficial outcomes were lost after one month [75]. EGCG at 42 mg/kg/day for one month rescued phosphoprotein deregulation in the hippocampus of Ts65Dn mice, reversing the kinome deregulation process and restoring the epigenetic profile, the beneficial effect in plasticity alterations being more beneficial if the treatment started during the first years of life [61,76].

Due to its pro-cognitive effects, EGCG therapy has therefore been proposed for DS. Another study has focused on another DS feature, the craniofacial abnormalities. Dysmorphic craniofacial phenotypes have been observed in humans with DS, these abnormalities taking place during prenatal morphogenesis and growth. Treatment with EGCG at 200 mg/kg was conducted by oral gavage of pregnant Ts65Dn mice and analyzed at E7 and E8, these mice exhibiting craniofacial dysmorphology as with newborns with DS. Some craniofacial corrections lasting until adulthood were seen, but with no effect with a lower dose [77]. Moreover, no improvements in male Ts65Dn trabecular bone and only limited improvements in cortical measures were observed after seven weeks of treatment with 20 mg/kg/day [78].

At the molecular effect, recent studies have focused on reduced synaptic function by the analysis on synaptic proteins, such receptors for glutamate and gamma-aminobutyric acid (GABA), or other neuromodulators, resulting in hippocampal disabilities which affect spatial memory, cognitive skills, and abilities. EGCG treatment also acts on hippocampal neurogenesis [79]. Given the role of DYRK1A on the brain defects in DS, many studies have analyzed the effect of inhibition of DYRK1A on learning and memory of mice models. Several compounds have inhibitory activity for DYRK1A. Among them, EGCG [80] has been used many times in preclinical studies, considering its capacity to inhibit DYRK1A activity with IC50 of 0.33 µM and the safety of the molecule [81]. Thus, multiple outcomes of EGCG in cognition related to DS have been possible through the inhibition of DYRK1A kinase activity.

Increased level of DYRK1A might affect GABAergic and glutamatergic pathways shifting excitation/inhibition neurotransmission balance toward an inhibition leading in fine to delayed cognitive progress. The first study to demonstrate the effect of DYRK1A inhibitors has used a DS mouse model carrying a YAC construct overexpressing Dyrk1A with other genes (Pipg, Ttc3, Dscr9, Dscr3). Green tea polyphenols treatment at 0.6 to 1.2 mg/day was initiated at the beginning of pregnancy until adulthood and rescued cognitive impairment and had a favorable impact on brain morphology, particularly in the thalamus-hypothalamus region, plus in object recognition memory [82]. Moreover, they found that BDNF levels, a neurotrophic factor known to be a major regulator of transmission and plasticity in the synapse in the adult brain, have been restored in the hippocampus of mice after treatment [71,82].

Using transgenic mice overexpressing Dyrk1A, it has been demonstrated that EGCG rescues behavioral phenotype due to the interconnection of Dyrk1A on GABAergic and glutamatergic signals [71,83]. Green tea extract (from LifeExtension For Longer Life, 45% EGCG) given in mice overexpressing Dyrk1A at 120–200 mg/kg/day for 4–6 weeks improved long term potentiation with the restoration of spine density [60]. Treatment for one month with EGCG-containing extracts (Polyphenon 60 at 225 mg/kg/day) restored excitatory/inhibitory (E/I) imbalance through the modulation of the GABA pathway through changes in the proteins involved in neuronal E/I balance, including GAD65, GAD67, enzyme metabolizing glutamate into GABA as well as VGAT1, a GABA transporter in Ts65dn mice [83]. This treatment was also able to improve corticohippocampal-dependent learning and working memory by rescuing dendritic spine density of the CA1 region of the hippocampus and normalizing the proportion of E/I synaptic markers in CA1 and dentate gyrus, thus improving [83].

The prenatal treatment of mice overexpressing Dyrk1A and the Dp(16)1Yey mouse model, which carries 113 genes orthologous to the genes on HSA21, until adulthood with a daily dose of 50 mg/kg in food pellets with Mega green tea extract showed a beneficial effect on inhibitory markers, on VGAT1/VGLUT1 balance, and density of GAD67 interneurons in the hippocampus, the consequence being an improvement of novel object recognition memory [84]. The beneficial effect was maintained in the animal up to the age of 69 days when the treatment was stopped at weaning, whereas treating adult mice did not show correction of these phenotypes [84]. Normalization of relevant plasma and neuronal biomarkers, such as BDNF and NFkB, was found after consumption of a diet enriched with EGCG (94% EGCG of total catechins) in mice overexpressing DYRK1A [71].

Although EGCG is widely known for its properties as an inhibitor toward DYRK1A kinase activity, other molecular processes of EGCG could also explain the pro-cognitive effect of green tea extract therapy. There are publications about oxidative stress and mitochondrial dysfunction being associated with DS. Brain mitochondria are known to be essential for many key processes, including energy production and redox homeostasis for neural development. Interestingly, Valenti and his colleagues have demonstrated an altered ATP synthesis associated with an excess level of production of ROS in the mitochondria of DS cells [67]. When the mitochondrial respiratory chain is not operative, it is the main producer of ROS. It has consequences in biogenesis, structure, dynamics, and metabolism of the mitochondrial of T21, which subsequently is associated with neurogenesis and neuroplasticity impairment [85]. In experiments using EGCG in cultured fibroblasts and lymphoblasts from DS patients, Valenti and his colleagues have been able to show a protective effect on mitochondrial energy deficit and oxidative stress by lessening mitochondrial energy damage. Their results show that EGCG has improved the performance of the mitochondrial oxidative phosphorylation by increasing cAMP level/PKA activity, ensuing the increase in NDUFS4 phosphorylation in favor of complex I activation and by increasing Sirt1 activity, a crucial regulator for promoting mitochondrial biogenesis in conditions of energy deficiency. Besides, their results indicate that EGCG enhances NRF-1 and T-FAM, proteins known to be implicated in the regulation of the expression of mtDNA encoded mitochondrial genes and thus inducing mitochondrial biogenesis [67]. In another study, EGCG appeared to have a positive effect on hippocampal neurogenesis through the improvement of mitochondrial bioenergetics and biogenesis in cultured neuronal progenitors’ cells from the Ts65Dn mouse model [68]. These in vitro findings can also be related to another in vivo study, Scala and coworkers have shown that in DS children aged between 1 and 8 years, a treatment of decaffeinated EGCG enriched by omega-3 for 6 months ensues the repairing of the deficit of mitochondrial complex I and ATP synthase. Moreover, they give evidence about the safety of EGCG in children without side effects in hepatic, renal, and thyroid function [85].

EGCG has then been proposed as a treatment of some DS phenotypic features. EGCG has been given in children, adolescents, and young adults with DS [72,85,86]. In all cases, EGCG is well tolerated [72,85,86]. Facial dysmorphology was improved in the Ts65Dn mouse model administered from embryonic day 9 to postnatal day 29 with green tea extracts at 30 mg/kg/day and observed in young children treated [87]. Two clinical trials in double-blind placebo-controlled were performed in DS individuals using EGCG treatment. In the first phase I study, a green tea extract with a standard quantity of EGCG was used. In a pilot study, 31 young adults with DS aged 14 to 29 years were enrolled in a randomized, double-blind study to test a green tea extract (oral dose of 9 mg/kg/day, Mega Green Tea Extract, Lightly Caffeinated, Life Extension^®^, Fort Lauderdale, FL, USA) or placebo treatments over a period of 3 months. EGCG treatment was shown to improve psychomotor speed and social functioning, visual recognition memory and working memory performance. An improvement of lipid profile, including total and LDL cholesterol, was found with no alteration of the hepatic function [72]. In a second clinical study, the safety and efficacy of cognitive training with a standard green tea extract supplementation versus cognitive training alone have been evaluated in a double-blind, randomized, placebo-controlled, phase 2 trial (TESDAD study) for 12 months (Life Extension Decaffeinated Mega Green Tea Extract; Life Extension^®^, Fort Lauderdale, FL, USA). Cognitive training alone or with EGCG supplementation was assigned to 84 young adults with DS aged from 16 to 34 years old. After that period a follow-up of the participants in the study was carried out over a further 6-month post-treatment period. For the combined cognitive training with EGCG treatment, improvements in executive functions, memory and adaptative behaviors were demonstrated. No changes have been observed in hepatic functionality. Lastly, improvements in functional connectivity and normalization of cortical excitability were seen by neuroimaging analysis when combined EGCG treatment and cognitive training [86]. By this study, the authors affirm a real effect of EGCG-cognitive-training in memory defect supported by a good safety and tolerability without major side effects in DS adults if the EGCG dose remains in low concentration. Moreover, the beneficial effects of green tea extract containing EGCG combined with cognitive training seem to be persistent 6 months after the end of the experiment [86]. The effects of this treatment were also analyzed on body composition and lipid metabolism, showing that EGCG has a sex-dependent effect on lipid profile that is related to changes in body mass and composition [88]. These findings reinforce therefore results obtained during the pilot study showing an improvement of visual recognition memory and adaptive behavior [72].

Despite these convincing findings, they are some studies stating the absent effects of EGCG in trisomic mice or the presence of adverse effects. Recent studies have reported that treatment with a low dose (10–20 mg/kg/day) or high dose (~50 mg/kg/day or 100 mg/kg/day) of a pure stabilized form of EGCG to Ts65Dn mice did not have any beneficial effect in cognitive or behavioral phenotype as observed in others study but inversely have harmful effects in skeletal structure in both trisomic and euploid mice [89,90,91]. In another study, Cuatuara and coworkers have shown that administration of EGCG (30 mg/kg/day) in 5–6 month aged Ts65Dn mice for 30 days did not have any effect on spatial learning [73]. Moreover, in Stagni et al. study, after one-month discontinuation, EGCG seems to have some side effects in both euploid and Ts65Dn mice [75]. Other results have shown that an incline in EGCG led to a decrease in plasma folate level in DS children and extra care about this criterion should be assumed. A clinical evaluation in DS children aged between 1 and 8 years has shown that treatment of decaffeinated EGCG supplemented by omega-3 for 6 months does not have any effect on their mental and psychomotor development compared to the DS control group [85].

In this respect, to have valuable long-term effects on DS-related cognitive impairment, forthcoming studies’ objectives are to fix a fitting EGCG dose, combined with or without other bioactive components (EGC, ECG and EC) or cognitive/social training and a proper critical time window to have valuable long-term effects on DS-related cognitive impairment. As well, understanding the interaction of EGCG with any other substance that induces degradation may be proper.

## 3. Alzheimer’s Disease and Catechins

### 3.1. Alzheimer’s Disease

Neurodegenerative diseases are a range of diseases of the brain, including AD, resulting in brain atrophy and neuron damage, and consequently the loss of cognitive or physical abilities. Elderly diabetes and stroke were linked with dementia, tea consumption being associated with a low prevalence of severe cognitive impairment and AD [92]. Amyloid proteins are performed from amyloid precursor protein (APP). Among various isoforms of amyloid β protein (Aβ), 1–40 and 1–42 are the most common, with 1–42 exhibiting the highest toxicity. The oligomerization and accumulation of Aβ are linked with cognitive decline and brain atrophy [93], with soluble Aβ oligomers producing neurotoxic effects [94]. Hyperphosphorylation of Tau, a protein related to microtubule structure and function, leads to aberrant protein aggregation resulting in dysfunction of axonal transport in AD [95]. Cholinergic and glutamatergic neurotransmitters are the main types involved in AD; glutamate can be increased by soluble Aβ oligomers [96].

Currently, the two major hallmarks, β-amyloid plaques, and neurofibrillary tangle formation, are due to phosphorylated tau protein, as well as microglia-triggered inflammation, changes in cholinergic neurotransmission, glutamate receptor over-activation, calcium homeostasis alterations, excessive generation of ROS and nitric oxide species, mitochondrial dysfunction, elevated pro-apoptotic protein expression, and synaptic dysfunction and loss [97,98,99,100]. Targeting these mechanisms may offer neuroprotection. A meta-analysis demonstrated that daily tea consumption is associated with a decrease in cognitive decline [101].

### 3.2. Catechins in Molecular Effects of Alzheimer’s Disease

The generation of ROS in neuronal cells activates proapoptotic mediators, such as Bcl-2 and caspase-9, as well as inflammatory mediators, such as TNF-α, COX-2, NF-κβ pathway [102]. Catechins may protect from neurodegenerative diseases by reducing the oxidative stress, thus decreasing directly or indirectly neuronal damages by improving antioxidant enzymes and scavenging ROS [103]. Pre-administration of EGCG in mice injected with lipopolysaccharide suppressed cytokines and inflammatory proteins increased levels thus prevented memory impairment [104]. In a streptozotocin-induced dementia model in the rat, decreased ROS levels in the hippocampus were associated with reversion of cognitive deficits assessed by the Morris water maze after oral administration of 10 mg/kg/day of EGCG for a month [26].

EGCG can protects against inflammation in AD linked with microglial activation [105]. Nrf2 signaling pathway can be stimulated by EC in primary cultures of astrocytes [106]. EGCG can activate cell survival pathways (PI3K/Akt, ERK and protein kinase C (PKC)), by inhibition of pro-apoptotic proteins and upregulation of anti-apoptotic and pro-survival genes [107]. EGCG can improve the restraint stress-induced neuronal impairments in rats accompanied by a partial restoration of normal plasma glucocorticoid, dopamine, and serotonin levels, PKCα expression and ERK1/2 phosphorylation [108]. Chronic EGCG treatment rescued learning and memory deficits in rats under chronic unpredictable mild stress, attenuating neuronal damage in the hippocampal CA1 region by a decrease in apoptotic cells, and reduction of soluble and insoluble Aβ1–42 levels in the hippocampal CA1 region of stressed rats [109].

Green tea catechins can also suppress morphologic and functional regression in the brain and memory regression in aged mice with accelerated senescence [110]. In this mice model that ingested EGCG at 20 mg/kg, the learning ability was significantly higher than that of mice that ingested either EGC or GA alone [111]. EGCG can therefore suppress cognitive dysfunction and may also protect neurons by activating cell survival signaling pathways [112]. In 4-month-old male senescence-accelerated mouse prone-8 (SAMP8) mice, a murine model characterized by early onset of learning and memory deficits along with spontaneous overproduction of soluble Aβ in the brain, spatial learning and memory impairments were prevented by oral administration of 0.05 or 0.1% green tea catechins in drinking water for 6 months [113].

Many studies have demonstrated the beneficial effect of green tea consumption in AD mice models [114]. Using EGCG at 2 mg/kg/day or 6 mg/kg/day for four weeks, neuronal injury and Aβ reduction in the hippocampus was shown [115]. Treadmill exercise combined with oral administration of EC at 50 mg/kg/day for four months reduced Aβ levels and suppressed tau protein aggregation with protection against cognitive deficits and in APP/PS1 mice [116]. APP/PS1 mice treated for four months with EGCG showed reduced microglia activation and Aβ plaques in the hippocampus with lower levels of IL-1β and higher levels of anti-inflammatory cytokines IL-10 and IL-13 [117].

Using the Swedish mutant APP (APPSw) transgenic AD mouse model, suppression of phosphorylated tau isoforms was found after intraperitoneal injection of 20 mg/kg EGCG for 60 days and 50 mg/kg EGCG orally treated for six months [118]. EGCG inhibits amyloid precursor protein cleavage and β-amyloid formation, reducing cerebral amyloidosis in mice [119]. α-secretase, which cleaves APP generating soluble APP-α (sAPP-α), can inhibit Aβ formation. Co-treatment with EGCG and fish oil, which enhances the bioavailability of EGCG, increases sAPP-α production and inhibits cerebral Aβ deposits in Tg2657 mice [120]. After 2 weeks of oral treatment with EGCG, PKC α and ε increase in the hippocampi of mice [121]. Taken together, these results underline the beneficial effects of catechins on PKC-related mechanisms by their role on soluble sAPP generation.

## 4. Metabolic Syndrome, Microbiota and Catechins

Metabolic syndrome is a concept that has been established in the early 2000s and that is defined by the presence of at least three of the five following criteria: waist circumference, high blood pressure, high blood glucose level, high triglycerides level and low HDL level. People with metabolic syndrome have a greater risk to develop cardiovascular diseases and/or type 2 diabetes (T2D). The underlying mechanisms that could explain metabolic syndrome are not yet elucidated but the most common explanation is that the energy homeostasis is unbalanced between the energy intake and the energy expenditure due mainly to a sedentary lifestyle: a lack of physical activity and an excessive calorie intake with a diet rich in lipids and carbohydrates and poor in fruits, vegetables, and fiber.

A well-controlled diet is a key component of diabetes management or prevention. While most studies focus on the number of calories or the type of fat, some studies have focused on micronutrients, such as polyphenols, and in particular, flavonoids.

EGCG has been shown to have multiple beneficial effects. EGCG isolated from green tea and injected I.P. daily reduced body weight, plasma insulin, plasma leptin and food intake in only 7 days in male and female Sprague Dawley rats and in obese Zucker rats, while epicatechin, ECG and EGC showed no effect [122]. EGCG administration decreased the fasting plasma level of free fatty acids and insulin by modulating diet-induced inflammation in adipose tissues of rats fed a high-fat diet [123]. EGCG may be beneficial in diabetes for increasing insulin sensitivity by suppressing inflammation [124].

However, green tea extract given in the diet to male Goto-Kakizaki rats, a model of non-insulin dependent diabetes, has no effect on food intake, body weight or plasma insulin, indicating that the purity and the type of catechin seem important as well as the method of administration to improve diabetes [125]. In a double-blind, placebo-controlled clinical study with obese patients without diabetes, EGCG was given twice a day for 8 weeks and has shown to lower plasma triglycerides without any changes in body weight, waist circumference or total body fat mass, and in vitro experiments on preadipocytes cell line exhibit no effect on lipolytic metabolism or browning [126] while EGCG given orally to HFD mice for 4 weeks induces an increase in brown adipose tissue thermogenesis [127].

EGCG, but not EC, can also have beneficial properties in diabetes management. Indeed, it has been shown in an in silico/in vitro study that EGCG can prevent amyloid aggregation of insulin, an occasional side effect of insulin injection, especially with the use of insulin pumps [128]. Catechins extracted from lychee seed (*Litchi chinensis*) has shown to improve insulin resistance by increasing glucose consumption rate and glycogen deposit in two insulin resistance cell lines models [129]. EGCG given during one month in T2D rats by oral gavage has been shown to decrease glucose plasma level and to improve insulin sensitivity (by decreasing HOMA-IR, an index of insulin resistance) and lipid profile [130]. The authors also show a cardioprotective effect of EGCG that seems to be driven by the anti-inflammatory and antioxidant effects of EGCG. Indeed, another flavonoid-rich extract (from *Abroma augusta*) has been shown to have similar cardioprotective effects in T2D rats by inhibiting oxidative stress [131]. Oxidative stress driven by hyperglycemia induces cardiomyocytes apoptosis, which is a hallmark of diabetic cardiomyopathy and leads to a decrease in cardiac contractile function [132]. Endoplasmic reticulum (ER) stress leads to an increase in unfold/misfold proteins that are linked to cardiomyopathy or heart failure [133] and hyperglycemia leads to an increase in ER stress [134,135]. In an in vitro study, Kim et al. have shown that several flavonoids (flavones, isoflavones and flavonols) have protective effects on ER stress in H9c2 cardiac muscle cells after ischemia/reperfusion by increasing Bcl-2 level, an anti-apoptotic protein and by decreasing Bax level, a proapoptotic protein and by decreasing the activation of caspase-3, a major protein involved in cell death [136]. Icariin [137], a purified flavonoid extracted from *Epimedium brevicornum* or silibinin [138], a purified flavonoid extracted from the seed of milk thistle (*Silybum marianum*) or artichoke (*Cynara scolymus*) have shown to have a similar effect on H9c2 cells by inhibiting ER stress. These in vitro results can be found also in an in vivo study, Zhao et al. has shown that in leptin-receptor deficient (db/db) mice, a mouse model of obesity and T2D, a berry powder of *Amelanchier alnifolia*, a common shrub found in western Canada, rich in anthocyanins, supplemented in a diet for 4 weeks, induces a decrease in ER stress markers in the heart and aorta, without change in body weight, food intake, or blood glucose of cholesterol levels, indicating the direct effect of its flavonoids content on cardiovascular health [138]. Db/db mice also present an encephalopathy, a major complication of diabetes, and quercetin, the most common food flavonoid mainly found in fruits, vegetables, or red wine [139], given orally for 12 weeks by gavage to db/db mice, has shown to improve learning and their memory ability by inhibiting the expression of ER stress signaling pathway [140]. Quercetin also induces an increase in Bcl-2 expression and a decrease in Bax and caspase-3 expression, indicating the decrease in neural apoptosis [141].

Among the “hot topics” of the past few years, gut microbiota is certainly one of the most important for its relevance and its implication on human health. Polyphenols can influence gut microbiota and have an impact on metabolic health. In germ-free mice, a fecal microbiota transplant from mice fed a high-fat, high sucrose (HFHS) diet supplemented with a proanthocyanidins or anthocyanins fraction from a powder of a blend of two species of blueberries (*Vaccinium ashei* and *Vaccinium corymbosum*), improves glucose homeostasis and in the case of proanthocyanidins fraction, increase locomotor activity [142]. Another research paper from the same team has shown that a polyphenol-rich cranberry extract (*Vaccinium macrocarpon*), mainly proanthocyanidins and flavonols, improves insulin sensitivity, lipid profiles and prevents weight gain in mice fed a HFHS diet by increasing the *Akkermansia* spp. population in the gut microbiota [143]. A polyphenol extract from blueberry (*Vaccinium* spp.) supplemented in mouse diets for 12 weeks also influences gut microbiota by modulating different species of bacteria, prevents diet-induced gain weight and normalizes plasma lipid profiles [144]. However, polyphenols do not have only an effect on gut microbiota, the opposite effect is also possible. Indeed, an in vitro study has shown that a 4–8h incubation of catechins and epicatechins with fecal microbiota can generate metabolites with more antioxidant properties, highlighting the importance of microbiota in the beneficial effect of catechin on metabolic health [145].

Short Chain Fatty Acid generated in the gut through reactions among undigested carbohydrates, catechins, and gut microbiota may enhance lipid metabolism through AMP-activated protein kinase (AMPK) activation, leading to their anti-obesity activity [146]. EGCG has been shown in vivo and in vitro to activate AMPK, an enzyme down-regulating lipogenesis and up-regulating lipolysis [147,148]. Pancreatic lipase can be inhibited in vitro by EGCG in a non-competitive manner, while EGC was ineffective [149,150].

In a randomized, placebo-controlled, double-blind study, a daily intake of anthocyanins extracted from European blueberry (*Vaccinium myrtillus*) and black currant (*Ribes nigrum*) in a diabetic population for 24 weeks, decreased plasma triglycerides by 23%, fasting plasma glucose by 8.5% and HOMA-IR by 13% [151]. However, even a single dose of polyphenols extracted from *Vaccinium myrtillus* (36% of anthocyanins) is sufficient to induce a decrease in plasma glucose level during an oral glucose tolerance test in T2D patients [152]. Some studies have focused on the mechanisms behind this decrease in plasma glucose level after flavonoid ingestion. In HFD mice, Dong et al. have found that a diet enriched with quercetin for 12 weeks can induce an increase in GLUT4 mRNA expression, a glucose transporter, in epididymal adipose tissue by activating the Akt signaling pathway [153]. However, quercetin can also have an upstream effect by inhibiting intestinal glucose uptake [154]. Flavonoids, such as quercetin [155], anthocyanins [156] or epicatechin [157], can also have protective effect at the pancreatic level, by preventing the beta cells from oxidative stress damage.

Flavonoids can also have an influence on food intake by having a direct effect at the brain level. EGCG, quercetin and, to a lower extent, anthocyanins, can pass the blood-brain barrier [158]. Leptin is the key hormone that regulates food consumption. Leptin resistance leads to an impaired leptin signaling pathway that results in an increase in food intake. Proanthocyanins, extracted from grape seed (*Vitis vinifera*), given orally to diet-induced obese rats for 21 days, show no effect on plasma leptin level, but have a beneficial impact on leptin signaling pathways in the hypothalamus, resulting in a decrease in food intake [159]. In a diet-induced obese mice model, EGCG has also a beneficial effect by reprogramming appetite-regulating signaling in the hypothalamus [160,161].

## 5. DYRK1A and Catechins

Preclinical and clinical trials have demonstrated the role of EGCG as inhibitor of Dyrk1A. However, no improvement of cognitive and behavioral phenotypes was seen after administration of pure EGCG at 9, 20 or 50 mg/kg/day to Ts65Dn mice [78,89,90,91]. Polyphenols of green tea extracts contain other catechins with bioactive potential effects in combination with EGCG on learning and memory in DS mice models. Interestingly, EGCG seems not to be the only catechin that has inhibitor effects on DYRK1A activity. In a previous study, we have analyzed the in vitro and in silico effects of green tea catechins not only for EGCG, but for others residually contained in a drink powder enriched with a standardized amount of EGCG (510 mg/100 g), because our pre-clinical findings demonstrate the safety of this drink which can also normalize relevant plasma and neuronal biomarkers deregulated in mice overexpressing Dyrk1A [71]. As with EGCG, we found that ECG was a noncompetitive inhibitor against ATP, this result being confirmed by molecular docking computations [140]. However, no effect was found for EGC and EC [71] and the role of CG and GCG to DYRK1A inhibition remains to be clarified.

We evaluated CG and GCG for their in vitro DYRK1A inhibitory activity using a fluorescent peptide substrate and UFLC (Ultra Liquid Chromatography) assay [71]. Table 1 shows Dyrk1A-ΔC (a recombinant 6xHis-tagged DYRK1A catalytic domain without C terminal sequence) activity and DYRK1A (DYRK1A native human protein, PV3785, Thermo Fisher Scientific, Waltham, MA, USA) activity, respectively, as a function of the concentration of each compound for 1 mM of ATP. The Dyrk1A-ΔC (a recombinant 6xHis-tagged DYRK1A catalytic domain) and Dyrk1A activities were found to decrease as the concentration of EGCG in the reaction increased (Table 1) [71]. The Dyrk1A-ΔC activity was also found to decrease in a dose-dependent manner with the concentration of CG and GCG (Table 1), the analysis of Dyrk1A activity confirming this result (Table 1).

We also analyzed the Dyrk1A-ΔC and Dyrk1A activities with each inhibitor with different concentrations of ATP (200–800 µM) and no difference was found (Table 2). Therefore, we determined that, as with EGGC, CG and GCG were noncompetitive against ATP.

To understand where and how the EGGC, CG and GCG interact with DYRK1A, a docking approach can be employed. This bioinformatics method aims to predict the preferred conformation and localization of a molecule, called the pose or binding mode, at the surface of a protein. To use docking methods, the tridimensional structure of the protein must be solved experimentally. A scoring function that counts the number of favorable intermolecular interactions is used to evaluate the different poses generated by the docking program. Usually, the pose with the lowest energy is the most probable. The docking of EGCG on the DYRK1A structure has been performed previously [71] using the Smina program [162] with a Vinardo scoring function [163]. The docking results have shown that EGCG binds preferentially in a flat pocket centered around L457, which is far from the kinase domain. Inside this pocket, EGCG forms a one cation-π interaction with K222 and three hydrogen bonds involving H424, R458 and Y462. The fact that the most probable EGCG pose is not in the catalytic site indicates that its inhibitory effects are not due to competition with ATP. Similar results were obtained for ECG.

The same protocol was applied for CG and GCG. CG also binds in the flat pocket around L457 with the same interactions than EGCG (Figure 3A). Surprisingly, according to the docking computations, GCG has best affinity for the catalytic site of DYRK1A and a lower affinity for the pocket around L457. These results disagree with the experimental results. Sometimes, the scoring function is not able to correctly evaluate the affinity between a ligand and a protein. Several reasons can explain this issue, such as a protein structure containing errors, false ligand parametrization, inadequate sampling efficiency of both ligand and/or receptor conformations, etc. Whatever the causes, based on the experimental results, we can discard the pose in the catalytic site and focus on the pose in the flat pocket around L457 (Figure 3B). The GCG binding mode in this pocket is slightly different to those of EGCG, EGC, and CG. The cation-π interaction with K222 and the two hydrogen bonds involving H424 and Y462 are still present. However, the GCG binding mode is characterized by the missing hydrogen bond with R458, which is replaced by a new hydrogen bond implicating the carbonyl carbon of the L377 backbone.

In conclusion, according to the docking results, the catechins that are noncompetitive inhibitors against ATP seem to target the same site on the DYRK1A surface. Further experimental and bioinformatics studies are now required to understand how, at the atomic level, these catechins disrupt the biological activity of DYRK1A.

## 6. Conclusions and Perspectives

Although DS is the main cause of intellectual disabilities, it is also characterized by a large set of comorbidities having developmental origins but also appear across an individual’s life span. Treatment with EGCG has been extensively shown in preclinical and clinical studies as an approach to manage DS targeting synaptic- and plasticity-related mechanisms involved in learning and memory to improve cognitive impairment. However, it will be necessary to study its effects on DS comorbidities. Dyrk1A is one of the main proteins implicated in cognitive impairment in DS, Dyrk1A inhibition emerging as a promising therapeutic pathway in DS. Interestingly, positive effects on adipogenesis in a cellular differentiation model [164,165] and on proliferation of pancreatic β-cells [165] have been showed by Dyrk1A inhibition. Moreover, recent data confirmed the interest in Dyrk1A inhibitors in a therapeutic approach of AD [166,167]. Even if the main studies on Dyrk1A inhibition by phenolic compounds focused mainly on EGCG, clarifying the contribution not only of EGCG but also of ECG, CG and GCG in vivo in intellectual disability and DS comorbidities needs further evidence. Moreover, determining the fitting dose is one of the main objectives for long term effects to avoid side effects previously demonstrated with very high doses of EGCG [168].

## Figures and Tables

**Figure 1 nutrients-14-02039-f001:**
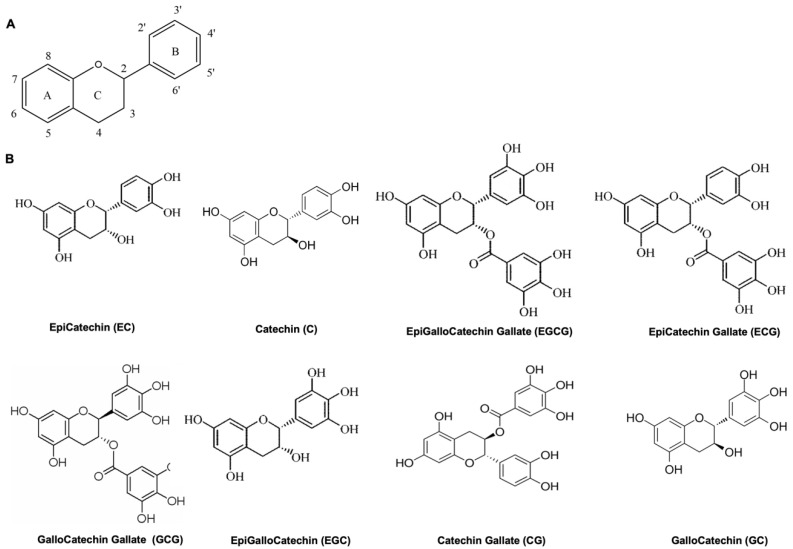
(**A**) Structure of the flavan nucleus, the basic structure of flavonoids. (**B**) Structure of eight catechins. Catechins have many chemical structural features, such as hydroxyl groups (–OH), that combine easily with other materials.

**Figure 2 nutrients-14-02039-f002:**
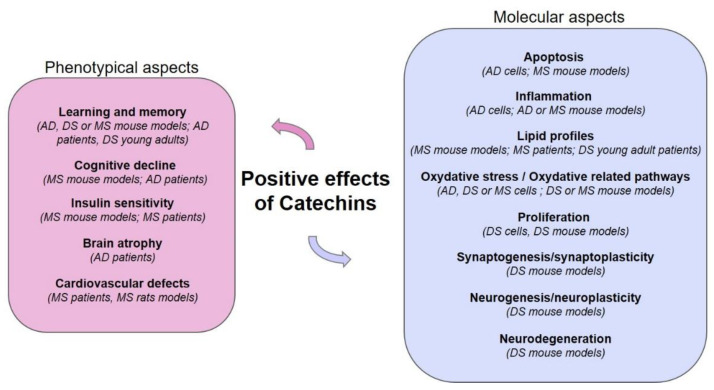
Major positive effects of catechins on several aspects of Down syndrome (DS) and its main comorbidities, Alzheimer’s disease (AD) and metabolic syndrome (MS).

**Figure 3 nutrients-14-02039-f003:**
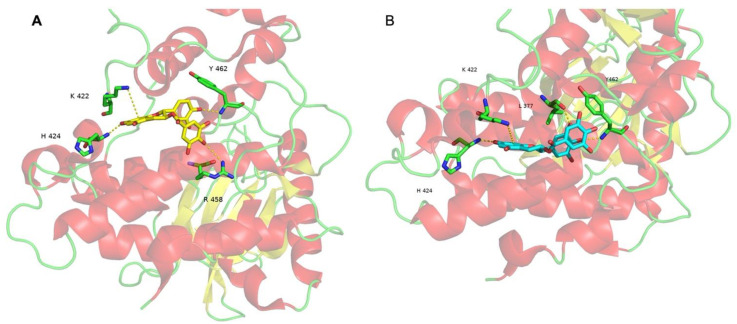
The docking results for (**A**) CG and (**B**) GCG. DYRK1A is in cartoon representation. CG, GCG and the residues forming non-covalent bonds are in sticks representation.

**Table 1 nutrients-14-02039-t001:** Dyrk1A-ΔC and Dyrk1A inhibition assays at different concentrations using HPLC-based assays.

Compound	Dyrk1A-ΔC Dyrk1A Remaining Activity at 0.1 μM (%)	Dyrk1A-ΔC Dyrk1A Remaining Activity at 1 μM (%)	Dyrk1A-ΔC Dyrk1A Remaining Activity at 10 μM (%)	Dyrk1A-ΔC Dyrk1A Remaining Activity at 100 μM (%)
CG	50.9 ± 4.4	5.7 ± 0.7	2.2 ± 0.2	0.9 ± 0.3
	(*n* = 3)	(*n* = 3)	(*n* = 3)	(*n* = 3)
	75.3 ± 13.3	5.1 ± 1	1.5 ± 0.2	0.6 ± 0.03
	(*n* = 3)	(*n* = 3)	(*n* = 3)	(*n* = 3)
EGCG	71.3 ± 4.6	16.3 ± 2.3	6.4 ± 1.1	3.5 ± 0.8
	(*n* = 13)	(*n* = 13)	(*n* = 13)	(*n* = 13)
	67.5 ± 9.4	10 ± 2	3.4 ± 0.6	2.1 ± 0.5
	(*n* = 7)	(*n* = 7)	(*n* = 7)	(*n* = 7)
GCG	48.2 ± 2.9	5.5 ± 0.6	3.2 ± 0.8	1.6 ± 0.4
	(*n* = 3)	(*n* = 3)	(*n* = 3)	(*n* = 3)
	78.2 ± 17.4	4.8 ± 1.5	1.4 ± 0.1	0.4 ± 0.1
	(*n* = 3)	(*n* = 3)	(*n* = 3)	(*n* = 3)

CG, Catechin Gallate; EGCG, EpiGalloCatechin Gallate; GCG, GalloCatechin Gallate. Inhibition of truncated DYRK1A (ΔDyrk1A) activity and Dyrk1A activity (normalized in the absence of inhibitor, expressed as percentage) obtained with different polyphenol concentrations (0.1–100 μM) and 1 mM of ATP and 60 μM of peptide as previously technically described [71].

**Table 2 nutrients-14-02039-t002:** Analysis of the inhibition of Dyrk1A-ΔC activity by CG, EGCG and GCG as a function of ATP concentrations.

Compound	Dyrk1A-ΔC Dyrk1A Remaining Activity with 200 μM ATP (%)	Dyrk1A-ΔC Dyrk1A Remaining Activity with 400 μM ATP (%)	Dyrk1A-ΔC Dyrk1A Remaining Activity with 800 μM ATP (%)
CG (10 μM)	6.1 ± 1.2	3 ± 0.5	3.5 ± 1.5
	(*n* = 4)	(*n* = 4)	(*n* = 4)
	2.5 ± 0.4	2 ± 0.3	1.7 ± 0.4
	(*n* = 4)	(*n* = 4)	(*n* = 4)
EGCG (10 μM)	11 ± 3.1	9.6 ± 2.1	5.4 ± 1.3
	(*n* = 4)	(*n* = 4)	(*n* = 4)
	6.3 ± 0.9	5.1 ± 0.9	4 ± 0.7
	(*n* = 4)	(*n* = 4)	(*n* = 4)
GCG (10 μM)	4.7 ± 0.8	4.1 ± 1.2	3.7 ± 0.9
	(*n* = 4)	(*n* = 4)	(*n* = 4)
	2.3 ± 0.4	1.7 ± 0.3	1.5 ± 0.3
	(*n* = 4)	(*n* = 4)	(*n* = 4)

Inhibition of truncated DYRK1A (ΔDyrk1A) activity and Dyrk1A activity obtained with 10 μM of CG, EGCG and GCG in the presence of different concentrations of ATP (200–800 μM) is shown. Assays were run for up to 60 min at 37 °C as previously technically described [71].

## Data Availability

Not applicable.

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
