# Peer review of "Catechins as a Potential Dietary Supplementation in Prevention of Comorbidities Linked with Down Syndrome"

_nutrients, 2022, doi:10.3390/nu14102039_

Round 1
Reviewer 1 Report
- The manuscript describes the role of catechins in Down syndrome in detail, and mentions their effects on Alzheimer’s disease and metabolic disorders. In the section Introduction, Lines 176-177, a final sentence, firstly mention Down syndrome. Therefore, it is better to reorganize the manuscript, i.e., Introduction immediately followed by “Down syndrome and catechins”, not “Alzheimer’s disease and catechins”.
- Introduction is too long. It should be concise.
- Correct many grammatical errors in the whole manuscript, such as line 27 “structure”, line 33 “Flavonoids”, line 37 “intestine”, and compound names in Figure 1B and lines 72-73.
- Rewrite lines 33-34, lines 73-74, lines 78-79.
- In figure 1A, remove the highlighted part.
- Line 109, give full name of NADPH
- Line 166, “absorbed by intestinal barriers” is wrong. Rewritten that.
Author Response
- The manuscript describes the role of catechins in Down syndrome in detail and mentions their effects on Alzheimer’s disease and metabolic disorders. In the section Introduction, Lines 176-177, a final sentence, firstly mention Down syndrome. Therefore, it is better to reorganize the manuscript, i.e., Introduction immediately followed by “Down syndrome and catechins”, not “Alzheimer’s disease and catechins”.
Answer: we have reorganized the manuscript and consequently changed the reference order.
- Introduction is too long. It should be concise.
Answer: we shortened the introduction (lines 25 to 117).
- Correct many grammatical errors in the whole manuscript, such as line 27 “structure”, line 33 “Flavonoids”, line 37 “intestine”, and compound names in Figure 1B and lines 72-73.
Answer: we carefully corrected the grammatical errors.
- Rewrite lines 33-34, lines 73-74, lines 78-79.
Answer: The sentences have been corrected (lines 32, lines 57-60, lines 61-64)
- In figure 1A, remove the highlighted part.
Answer: the figure has been corrected as requested
- Line 109, give full name of NADPH
Answer: this has been corrected (line 74)
- Line 166, “absorbed by intestinal barriers” is wrong. Rewritten that.
Answer: this has been corrected (line 108)
Reviewer 2 Report
It is a very well written review describing the various comorbidities associated with Down Syndrome and detailing how catechins can play a role in the various comorbidities and thus can overall affect the health and lifespan of people with Down Syndrome. Some suggestions to the authors to make the review more valuable to the readers who are interested in the biological understanding of the effects of catechin.
- Since the title is focused on Down syndrome comorbidities and not metabolic disorders or Alzheimer's disease in general it would be better to move the Down syndrome section at first and then discuss the role of catechins in Alzheimer's and metabolic conditions.
- The review can include more figures to again bring the focus on Down syndrome related conditions make it more appealing to the readers such as a representation of the different comorbidities associated with Down syndrome.
- Addition of a table to briefly describe the major effects of catechin in different conditions associated with Down Syndrome will provide a nice summary of the key points of the manuscript.
Author Response
- Since the title is focused on Down syndrome comorbidities and not metabolic disorders or Alzheimer's disease in general it would be better to move the Down syndrome section at first and then discuss the role of catechins in Alzheimer's and metabolic conditions.
Answer: we have reorganized the manuscript and consequently changed the reference order.
- The review can include more figures to again bring the focus on Down syndrome related conditions make it more appealing to the readers such as a representation of the different comorbidities associated with Down syndrome.
- Addition of a table to briefly describe the major effects of catechin in different conditions associated with Down Syndrome will provide a nice summary of the key points of the manuscript.
Answer: to answer points 2 and 3, we have added a figure (figure 2) which summarizes major positive effects of catechins not only on several aspects of Down syndrome but also for its main comorbidities, Alzheimer’s disease, and metabolic syndrome.
Round 2
Reviewer 1 Report
The manuscript is well re-organized.
Author Response
thank you
